# Engineering Metal–Organic Frameworks (MOFs) for Controlled Delivery of Physiological Gaseous Transmitters

**DOI:** 10.3390/nano10061134

**Published:** 2020-06-08

**Authors:** Mengdan Zhang, Ruirui Qiao, Jinming Hu

**Affiliations:** 1CAS Key Laboratory of Soft Matter Chemistry, Hefei National Laboratory for Physical Science at the Microscale, Department of Polymer Science and Engineering, University of Science and Technology of China, Hefei 230026, China; zmd95@mail.ustc.edu.cn; 2ARC Centre of Excellence in Convergent Bio-Nano Science and Technology, Australian Institute for Bioengineering and Nanotechnology, The University of Queensland, Brisbane, QLD 4072, Australia

**Keywords:** metal–organic frameworks, nitric oxide, carbon monoxide, hydrogen sulfide

## Abstract

Metal–organic frameworks (MOFs) comprising metal ions or clusters coordinated to organic ligands have become a class of emerging materials in the field of biomedical research due to their bespoke compositions, highly porous nanostructures, large surface areas, good biocompatibility, etc. So far, many MOFs have been developed for imaging and therapy purposes. The unique porous nanostructures render it possible to adsorb and store various substances, especially for gaseous molecules, which is rather challenging for other types of delivery vectors. In this review, we mainly focus on the recent development of MOFs for controlled release of three gaseous transmitters, namely, nitric oxide (NO), carbon monoxide (CO), and hydrogen sulfide (H_2_S). Although these gaseous molecules have been known as air pollutants for a long time, much evidence has been uncovered regarding their important physiological functions as signaling molecules. These signaling molecules could be either physically absorbed onto or covalently linked to MOFs, allowing for the release of loaded signaling molecules in a spontaneous or controlled manner. We highlight the designing concept by selective examples and display their potential applications in many fields such as cancer therapy, wound healing, and anti-inflammation. We hope more effort could be devoted to this emerging fields to develop signaling molecule-releasing MOFs with practical applications.

## 1. Introduction

Metal–organic frameworks (MOFs) are hybrid materials composed of metal ions or metal clusters and organic ligands. They have high porosity, large specific surface area, and surface tailorability. Since their discovery, MOF materials have found extensive application in gas storage [1,2,3] and separation [4], adsorption [5,6,7,8], catalysis [9,10], electromagnetic materials [11,12,13], and so on (Figure 1). In recent years, MOFs have gained increasing attention in the field of biomedical applications such as drug carrier, biological imaging and sensing, and theranostic nanovectors due to their unique advantages of large pore size, degradability, and adjustable size [14,15].

On the other hand, gaseous transmitters are endogenously generated gases that can freely penetrate cell membranes [16]. These signaling molecules have specific cell and molecular targets and possess important physiological functions. The three gaseous transmitters identified so far are nitric oxide (NO), carbon monoxide (CO), and hydrogen sulfide (H_2_S). All of them have multiple roles in physiological systems, which are of crucial importance in maintaining normal physiological functions. Moreover, these signaling molecules have shown unprecedented therapeutic capacity in the treatment of many diseases such as cancer, inflammation, and bacterial infection. However, the biological functions of gaseous transmitters are highly concentration-dependent. It is inconvenient to directly administrate the gas molecules due to the difficulty to precisely control the dosage. To attenuate the side effects of the systemic toxicity of these gaseous transmitters, the development of delivery platforms capable of delivering these signaling molecules to the pathological tissues and realizing on-demand controlled release is highly desirable. To date, small molecule-based donors and various nanocarriers including inorganic nanoparticles, polymer assemblies, biomacromolecules, etc. have been extensively studied. Although the porous structures and ultrahigh surface areas endow MOF materials with the intrinsic properties to store and release gaseous molecules, the development of MOF-based nanomaterials for controlled delivery has not receive much attention until recently.

In this review, we summarize the recent advances of MOF-based materials for the controlled delivery of gaseous transmitters (Appendix A). Specifically, MOFs for controlled delivery of NO, CO, and H_2_S are briefly discussed with selective literature reports. We pay special attention to the design concepts and show the biomedical applications in cancer therapy, anti-inflammation, wound healing, etc. In addition, we shed light on the future direction of this emerging field. The intention is not to provide an exhaustive literature survey but to show the many possibilities to develop MOF-based materials for controlled delivery of gaseous transmitters. We hope this emerging field could be of potential interest to more researchers and will bring more innovative ideas to advance this intriguing field.

## 2. MOFs for the Delivery of Nitric Oxide

NO, in addition to being an important chemical intermediate, has a free radical property which makes it play an important role in the physiological regulation of organisms. In 1998, Furchgott, Murad, and Ignarro were awarded the Nobel Prize in Physiology and Medicine for their seminal work of NO which unravels the physiological functions of NO. NO is endogenously produced by the stoichiometric conversion of L-arginine to L-citrulline in the presence of nitric oxide synthases (NOSs), and it plays an essential role as a messenger molecule in the body and participates in the regulation of physiological processes [17]. For example, NO is an important regulator and mediator in the nervous, immune, and cardiovascular systems. Furthermore, it could be used as a therapeutic agent for the treatment of cancer, bacterial infection, wounds, etc. [18,19]. At ambient condition, NO is a gaseous molecule and highly unstable in the air. This intrinsic property makes it difficult to directly administrate NO. Moreover, the physiological functions of NO are highly concentration-dependent. For example, although a high NO concentration (µM to mM) could efficiently inhibit the proliferation of cancer cells, a decreased NO concentration (<pM to nM) could be beneficial for cancer cell growth (Figure 2). As such, it is of crucial importance to develop NO-releasing materials capable of delivering NO to specific pathological sites and releasing NO on-demand with a proper concentration. In this regard, much effort has been dedicated to various NO-releasing small molecules and nanovectors, including organic-inorganic hybrid materials, macromolecular assemblies, and so on [20,21]. Recently, with the rapid development of MOFs, the high porosity, large surface area, and easy structural adjustment make them promising candidates for NO delivery [22]. To date, three main strategies have been proposed to develop NO-releasing MOFs: (1) coordination of the metal ions with NO, (2) functionalization of the organic ligands with NO-releasing moieties, and (3) MOF-catalyzed decomposition of conventional NO donors. These methodologies took full advantage of either the metal ions, the organic ligands of MOFs, or the combination of them. These emerging NO-releasing cargos have shown promising applications in many pathological models.

### 2.1. Coordination of the Metal Ions with NO within MOFs

It is well-documented that metal nitrosyl has been widely used as a sort of NO donor. For example, sodium nitroprusside (SNP) has been clinically used for hypertensive emergency. NO could efficiently coordinate to many metal ions such as ferric (Fe^2+^) and ruthenium (Ru^2+^) with the formation of metal nitrosyl complexes. Therefore, it is not surprising that metal ion-containing MOFs could be potentially used as NO donors through the formation of metal nitrosyls. For example, Pinto et al. [24]. designed a titanium (Ti)-based MOF material, MIP-177, forming nitrites on the MOFs skeleton. This Ti-based MOF exhibited a high NO storage capacity (3 µmol/mg solid, ranking as one of the best NO-loading materials [25,26]) and good biocompatibility (Figure 3). In biological media, it can be stably existed and slowly release NO. Moreover, the role of MIP-177 in wound recovery and skin repair was studied, and the biological response was highly correlated to the NO concentration.

CPO-27 was a kind of MOF material prepared by the coordination of 2,5-dihydroxyterephthalic acid with different metal ions and the unsaturated metal atom in the structure could further coordinate with NO [25]. Among CPO-27 (Ni), CPO-27 (Zn) and CPO-27 (Mg), CPO-27 (Ni) has the highest NO storage capacity [27]. However, CPO-27(Ni) was also greater in cytotoxicity than that of other counterparts. To circumvent this problem, CPO-27 (Zn) or CPO-27 (Mg) were doped with Ni to obtain hybrid MOFs with low toxicity and high NO-loading capacity [28]. Interestingly, the amount of NO released from Ni-doped MOFs was dependent on the doped Ni content, and the rate of arterial relaxation was positive with the Ni content as well. This work provided an excellent example to fabricate NO-releasing MOFs with high performance by integrating the strengths of two MOF materials, avoiding the use of ingredients with serious physiological hazards.

To further assess the toxicity of various MOFs, the toxicity of 16 nanoMOFs against HepG2 and MCF7 cell lines and zebrafish embryos was evaluated. The results revealed that the toxicity was mainly due to the leaching of metal ions from the MOFs. Among various metal ions such as Fe^2+^, Zn^2+^, Zr^2+^, Mn^2+^, Co^2+^, Ni^2+^, Cu^2+^, and Mg^2+^, Mg-based MOFs had the lowest toxicity [29]. In addition to screening of metal ions with low toxicity, the use of biocompatible ligands has also been investigated. In this context, Pinto et al. [26] designed a NO-releasing MOF based on vitamin B_3_ as the ligand. The metal ions of Ni^2+^ and Co^2+^ within the MOFs could further bind NO, allowing for the release of 2.6 and 2.0 µmol/mg NO for the Ni- and Co-based MOFs, respectively. Cytotoxicity tests revealed that these two MOFs exhibited negligible toxicity at concentrations below 180 µg/mL.

Apart from the evaluation of the NO-releasing MOFs in cells, the further assessment of these NO-releasing materials in vivo is more appealing. Diabetic wounds are one of the most serious complications of diabetes, which are slow to heal or even do not heal. Studies have shown that the slow healing of diabetic wounds is mainly caused by the high blood sugar content that inhibits the production of endogenous NO [30,31,32,33]. As a result, NO could be potentially applied as an emerging therapeutic drug for the treatment of diabetic wounds. In this context, Xu et al. [34] designed a MOF carrier by encapsulating NO-loaded HKUST-1 (NO@HKUST-1) within the hydrophobic polycaprolactone (PCL) matrix, followed by coating with hydrophilic gelation to avoid premature NO release (Figure 4). In the treatment of diabetes wounds, the hydrolysis of the MOFs resulted in the continuous release NO and Cu^2+^ ions at relatively low concentrations. Notably, the corelease of NO and Cu^2+^ ions exerted a synergistic effect on the treatment of diabetic wound through promoting angiogenesis and inhibiting wound inflammation.

Notably, the use of metal ions of MOFs capable of coordinating NO represents a new approach to fabricate NO-releasing materials. However, these NO-releasing MOFs typically suffer from insufficient stability in aqueous solution [35]. This limitation may impede the potential applications in a physiological condition. On the other hand, the toxicity of metal ions is another concern. For example, ferrous (Fe^2+^) ions may amplify intracellular oxidative stress through the Fenton reaction mechanism, whereas other heavy metal ions such as Cu^2+^ have intrinsic cytotoxicity [36].

### 2.2. Functionalization of the Organic Ligands of MOFs for Controlled NO Release

In addition to the metal ions, the organic ligands of MOFs could also be tailored by the installation of NO-releasing moieties. Cohen et al. [37] investigated that aniline-containing MOF materials which can be postmodified with NO to obtain diazeniumdiolate (NONOate)-functionalized MOF. Under a high NO pressure (100 psi, 24 h), two different MOFs containing aniline groups (IRMOF-3 and UMCM-1-NH_2_) were synthesized, IRMOF-3–NONO and UMCM-1–NONO. These two MOFs quickly decomposed in phosphate buffer solution (PBS) at pH 7.4 due to the poor stability of NONOate moieties. Although the stability of the NONOate functional group in these materials is limited, this work provided a feasible strategy to fabricate NO-releasing MOFs with amine pendants. In another contribution, Cu-based MOFs with NONOate as NO donor molecules were prepared and the secondary amine structure on the Cu-TDPAT organic ligand (H_6_TDPAT = 2,4,6-tris(3,5-dicarboxylphenylamino)-1,3,5-triazine) were modified with the formation of NO-releasing NONOate moieties [38]. Notably, the formation of NONOate moieties could remarkably elevate the NO-loading amount, which was 5-time higher in loading contents than for MOFs fabricated through the physical adsorption. This NO-releasing MOF was relatively stable in a dry condition at room temperature, but sustained NO release could be achieved within 7 days at a humidity of 85% at 37 °C.

In the above examples, the NO release from MOF materials was remarkably affected by moisture, resulting in premature NO leakage in physiological conditions. To address this issue, photo-responsive MOFs were developed to achieved controlled NO release. Furukawa et al. [39] devised a photoactive porous coordination polymer platform for spatiotemporally controlled release of NO. Briefly, imidazole-based ligands, 2-nitroimidazole (2nIm) and 5-methyl-4-nitroimidazole (mnIm), were applied to construct zeolitic imidazolate frameworks (ZIFs), NOF-1 and NOF-2 (Figure 5). Using these photo-responsive ligands enabled photo-mediated NO release under light irradiation. For example, under mild light irradiation (300 W Xenon lamp, 7.5 mW/cm^2^), the NO release of NOF-1 and NOF-2 increased significantly compared to the corresponding precursors, and the NO release contents were calculated to be 3.4 and 2.9 µmol/mg NO, respectively. This result was tentatively ascribed to the high light-capturing capacity of the photo-responsive ligands in the MOFs, and the inherent voids provided spatial isolation between donors, thereby preventing aggregation-induced quenching of reactive excitation species. Moreover, the NO release could be manipulated within cells by taking advantage of the spatiotemporal control of light stimulus.

Although the introduction of photo-responsive ligands rendered it possible to achieve controlled NO release, the stability in aqueous solution remained limited. To increase the stability of photo-responsive MOFs, Kitagawa et al. [40] designed a MOF material based on bis-*N*-nitroso (BNN) moieties and prepared NOF-11 and NOF-12 (Figure 6), which could release NO under white light irradiation (300–600 nm, 0.3 mW/cm^2^). The maximum NO release of NOF-11 and NOF-12 was estimated to be 1.60 and 2.78 mmol/g, respectively. Moreover, a detailed study revealed that the Ti-based MOF (NOF-11) showed the highest stability even in aqueous media, exhibiting promising application in a biological condition.

The combination of therapeutic and imaging modalities into one platform led to the formation of the so-called theranostic platform. Apart from NO delivery capacity, the incorporation of imaging modality into NO-releasing MOFs rendered it possible to in situ evaluate the therapeutic outcomes of NO. In this regard, Yin et al. [41] designed a multimodal MOF material for NO delivery, integrating imaging and photothermal therapy (PTT) modalities. In this contribution, Mn^2+^-porphyrin NMOFs were prepared and then modified with thermo-responsive S-nitrosothiol (SNO) to obtain NMOF-SNO. The Mn^2+^ ions could be used for magnetic resonance (MR) imaging while the porphyrin-contained MOFs doped with SNO residues could be potentially used for PTT with the release of NO upon light irradiation. In vivo antitumor study suggested that, under near-infrared (NIR) irradiation, the growth of tumors was significantly inhibited by the treatment of NMOF-SNO. Moreover, in a multidrug resistance (MDR) tumor model, NMOF-SNO showed better tumor suppressing efficiency than doxorubicin, a well-known anticancer drug. This result consolidated the synergistic therapy of PTT and NO, proving an efficient method to treat MDR tumors.

### 2.3. MOF-Catalyzed In Situ NO Generation

As discussed above, NO-releasing MOFs could be successfully fabricated through the modification of either metal ions or organic ligands. MOF materials have been extensively used for catalysts due to their unique porous structures and the presence of metal ions [9,10,42]. Interestingly, NO-releasing MOFs could also be fabricated by taking advantage of the catalytic performance of MOFs, which can trigger the decomposition of other NO donors, allowing for NO release in an indirect way. In this aspect, Cu-based MOFs such as Cu_3_(BTC)_2_ (BTC: 1,3,5-benzenetricarboxylate) [43] and CuBTTri (H_3_BTTri: 1,3,5-tris[1H-1,2,3-triazol-5-yl]benzene) [44] have been developed, generating NO by catalyzing the decomposition of SNOs [45]. Poly(vinyl alcohol) (PVA) has been widely used in biological applications due to the hydrophilicity and excellent biocompatibility. In addition, the incorporation of NO-releasing materials into PVA matrix has been studied for vasodilation, wound treatment, and cardiovascular diseases [46,47,48]. In a recent contribution, Reynolds et al. [49] incorporated Cu-BTTri into a hydrophilic PVA membrane, in which Cu-BTTri could promote the release of NO upon contacting endogenous GSNO. Notably, the dispersion of MOFs in the PVA matrix remarkably elevated the stability, which did not adversely affect the MOF-facilitated NO release from endogenous GSNO. Similarly, Weng et al. [50] prepared Cu-based MOFs deposited onto the surface of titanium metal by a layer-by-layer method [51,52], which can catalyze endogenous NO donors such as GSNO to produce NO as well.

As discussed above, MOF materials have received increasing attention in recent years due to the potential applications in many fields. However, many MOFs suffer from poor stability in aqueous solutions/physiological environments. Although the degradable property is beneficial for the biological metabolism of MOFs, the instability of MOFs may lead to the premature release of encapsulants. Moreover, the degraded products such as heavy metal ions and organic ligands may be detrimental for biomedical applications. As such, much effort has been devoted to improve the stability of MOFs in aqueous solutions. Serre et al. [53] recently summarized the preparation of stable MOFs using high-valence main group and transition metals of Al^3+^, Fe^3+^, Zr^4+^ and Ti^4+^ and ligands. Apart from the increased binding strengths between metal ions and organic ligands, another popular strategy involves the separation of MOFs from moisture by embedding water-labile MOFs into polymeric matrices [39]. To further advance the practical applications of these NO-releasing MOFs, it should carefully balance the stability and NO-releasing profiles to achieve targeted therapeutic benefits.

## 3. MOFs for the Delivery of Carbon Monoxide

Like NO, CO has been recognized as a signaling molecule. The endogenous CO production mainly depends on the breakdown of heme catalyzed by heme oxygenase [54,55]. CO exists widely in the human body, and plays a regulatory role on the respiratory system, cardiovascular system, nervous system, etc. [56,57]. Akin to the development of NO donors, a variety of CO-releasing molecules (CORMs) have been developed to increase the bioavailability and decrease the systemic toxicity of CO [58,59,60,61]. Although CO could readily bind to many transition metals, conventional metal carbonyl-based CORMs suffer from several limitations such as insufficient stability in a biological condition, spontaneous release, and limited action time. To this end, much effort has been devoted to the development of metal-free CO donors and some successful examples have been reported, including 3-hydroxyflavone (3-HF) derivatives [62], xanthene-9-carboxylic acid [63], boron dipyrromethene (BODIPY) [64], and aromatic α-diketone derivatives [65]. Despite tremendous achievements, the development of CO-releasing MOFs is rather sparse. Based on the results of iron carboxylate MOFs with good biocompatibility [14,66], Metzler-Nolte et al. [67] reported the preparation of iron-based MIL-88B-Fe and amino-functionalized NH_2_-MIL-88B-Fe using ferric(III) chloride hexahydrate and terephthalic acid and 2-aminoterephthalic acid. CO could bind to the unsaturated coordination sites within the MOFs and the CO release was achieved by the degradation of the MOF materials under physiological conditions.

To achieve controlled CO release within MOFs, Furukawa et al. [68] loaded a CO donor of MnBr-(dmbpy)(CO)_3_ onto the surface of the MOF of UiO-67-bpy through covalent coordination, resulting in the formation of a photosensitive CO-releasing framework (CORF-1) (Figure 7). Compared with the MnBr-(dmbpy)(CO)_3_ small molecule precursor, under identical 460 nm irradiation conditions, CORF-1 could quickly release CO, while no CO release was found for the MnBr-(dmbpy)(CO)_3_ precursor. Further investigation revealed that MOF materials provided abundant space sites for photosensitive CO donor molecules, which can effectively suppress the aggregation-induced quenching phenomenon, thereby improving the light absorption efficiency of photosensitive molecules and releasing CO.

## 4. MOFs for the Delivery of Hydrogen Sulfide

H_2_S is the third identified gaseous transmitter. Endogenous H_2_S is mainly produced by the action of three enzymes on the L-cysteine substrate, which are cystathionine β-synthase (CBS), cystathionine γ-lyse (CSE), and 3-mercaptosulfurtransferase (MST) [69]. The biological roles of H_2_S include regulating blood pressure, affecting inflammation, preventing apoptosis, and maintaining vascular tone [70,71].

Notably, although many MOF materials have been used for H_2_S study, many of them focused on the removal of this odorous air pollutant. So far, there are very few studies on the controlled H_2_S release using MOF-based delivery platforms. Due to the high adsorption capacity and excellent delivery performance of CPO-27, Morris et al. [72] prepared two MOFs (Ni-CPO-27 and Zn-CPO-27) based on Ni, Zn and 2,5-dihydroxy-1,4-phthalic acid ligands for the storage and delivery of H_2_S (Figure 8). Powder X-ray diffraction (XRD) was used to study the interaction of Ni-CPO-27 and H_2_S, and it was concluded that the interaction between metal atoms and sulfur was the main interaction of the system for gas storage. The H_2_S-loaded CPO-27 materials were placed in a humid environment to study the release of H_2_S. Ni-CPO-27 showed higher H_2_S loading capacity than Zn-CPO-27. The Ni-CPO-27 showed no degradation of the crystal structure within 6 months. In contrast, under the same conditions, although the crystal structure of the Zn-CPO-27 material was damaged, it still had the ability to deliver a large amount of H_2_S. Taking into account of the toxicological characteristics of Ni-CPO-27, the H_2_S-loaded Zn-CPO-27 was further studied for the release of H_2_S under physiological conditions, and it was suggested that the H_2_S released from the MOF materials did have biological activity involving vasodilation of the porcine artery.

## 5. Summary and Outlooks

In this review, we briefly introduced the recent achievements of MOF materials for the delivery of three gaseous transmitters including NO, CO, and H_2_S. The combination of the unique properties of MOFs and the therapeutic functions of gaseous transmitters provides many opportunities to develop novel delivery materials of signaling molecules. Although it remains at the infant stage of this emerging field, a variety of encouraging results have been achieved in the treatment of various diseases such as cancer therapy, wound healing, bacterial infection. Three strategies have been successfully used to fabricate NO-releasing MOFs, while only little work has been done in the field of MOFs for delivery of CO and H_2_S. There is still plenty of room in this emerging area.

First, the stability of MOFs remains a grand challenge and the leakage of metal ions, in some occasions, leads to severe toxicity. Although some attempts have been made to elevate the stability of MOFs, only a few successful examples have been achieved. In addition to metal ions, the biosafety of organic ligand should be prudently assessed as well. Notably, the incorporation of biocompatible ligands into the design of MOFs provides an efficient way to decrease the toxicity. The increased stability and decreased cytotoxicity would be a prerequisite for practical applications of MOF-based materials.

Second, it can be speculated that the three gaseous transmitters are related to each other and affect each other. Current studies mainly focus on the delivery of a single signaling molecule, and the development of MOFs for synergistic delivery of multiple signaling molecules could provide additional insights into the physiological functions of gaseous transmitters. Perhaps in addition to NO, CO, and H_2_S, there may be other gaseous transmitters in the body, such as ammonia (NH_3_), formaldehyde, sulfur dioxide (SO_2_), and so on. Therefore, more research work is needed in the future to focus on the restriction and synergy among these gas signal molecules.

Finally, some transition metal ions and organic ligands show great potential in fluorescence or MR imaging. The incorporation of these metal ions and (or) organic ligands renders the resulting gaseous transmitter-releasing MOFs with a unique imaging performance. As such, the therapeutic outcomes of gaseous transmitters could be in situ reported by fluorescence or MR imaging. Overall, the controlled delivery of gaseous transmitters using MOF materials as the carriers provides a robust strategy to integrate the therapeutic potential of gaseous transmitters and the unique physiochemical properties of MOFs. This emerging field is full of challenges as well as opportunities. We hope more innovative ideas appear to advance the practical application of these MOF-based smart materials.

## Figures and Tables

**Figure 1 nanomaterials-10-01134-f001:**
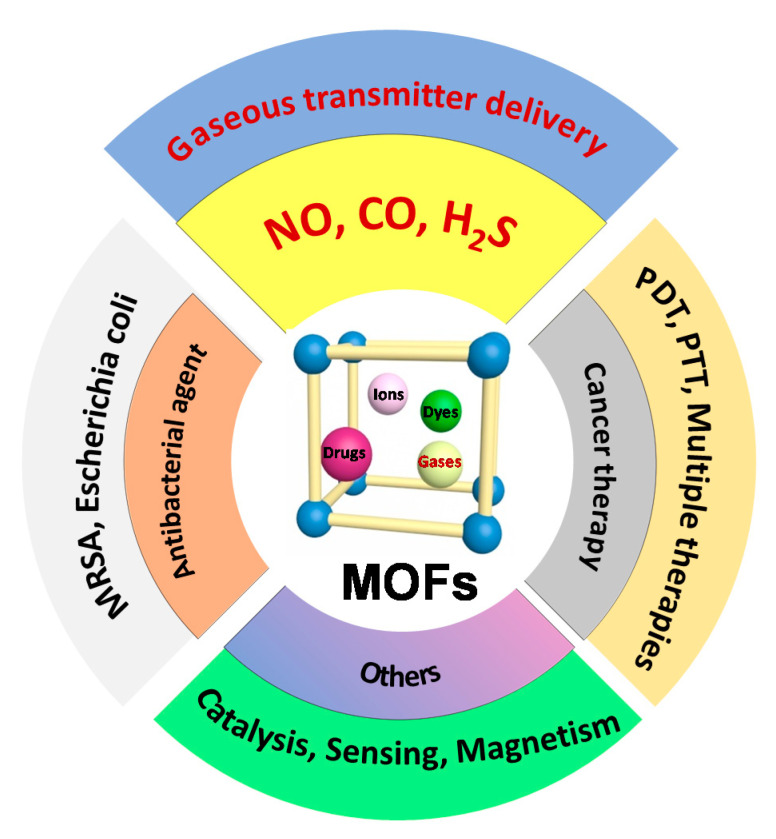
Functional applications of metal–organic framework (MOF)-based materials in diverse fields such as cancer therapy, antibacterial application, and delivery of gaseous transmitters.

**Figure 2 nanomaterials-10-01134-f002:**
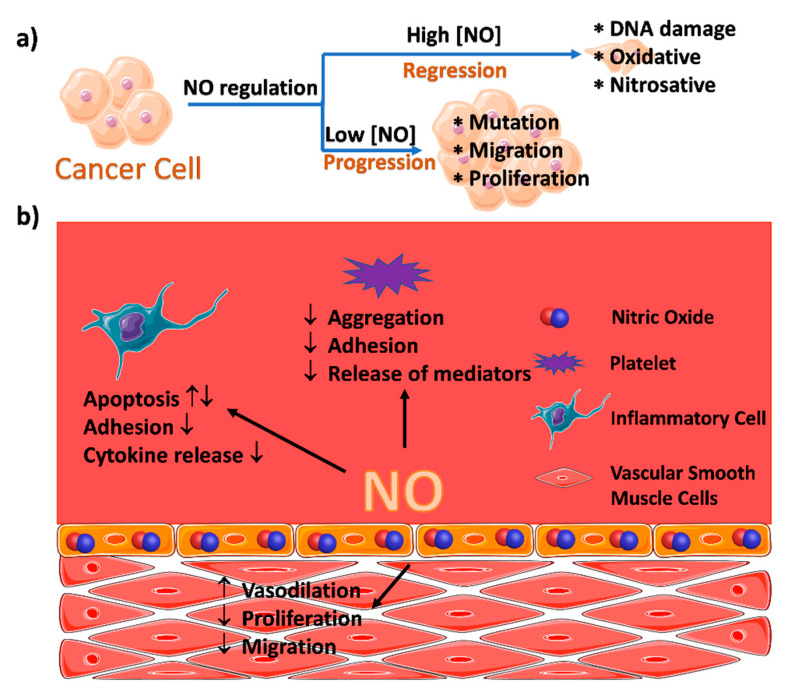
(**a**) The dual roles of NO in cancer biology. (**b**) The roles of NO in the vascular endothelium and its effects on cellular activities. Reproduced with permission from [23], copyright 2012, the Royal Society of Chemistry.

**Figure 3 nanomaterials-10-01134-f003:**
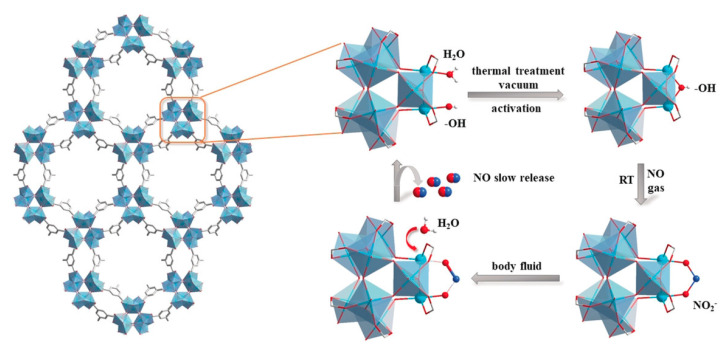
MIP-177 microporosity and binding/release mechanism: viewed along the *c*-axis (**left**), adsorption and controlled release cycle of NO under the tested conditions (**right**) is shown. Reproduced with permission from [24], copyright 2020, John Wiley & Son.

**Figure 4 nanomaterials-10-01134-f004:**
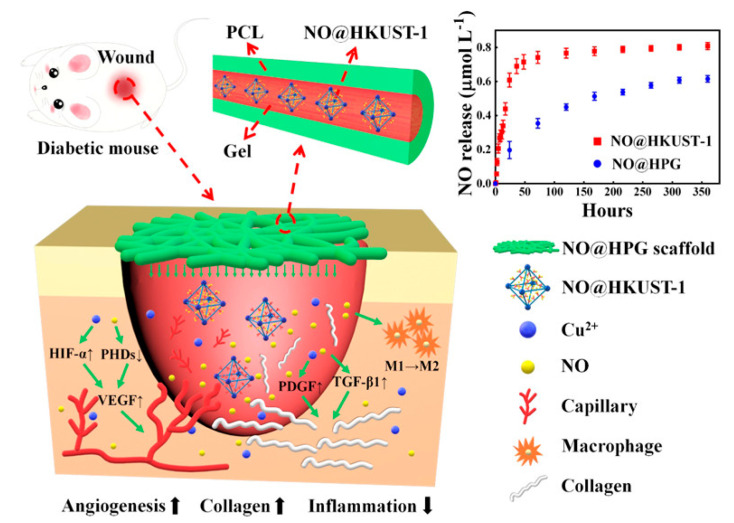
Schematic illustration of NO@HKUST-1 that spontaneously released NO and Cu^2+^ ions over time, facilitating the healing of diabetic wounds. Reproduced with permission from [34]. Copyright 2020, American Chemical Society.

**Figure 5 nanomaterials-10-01134-f005:**
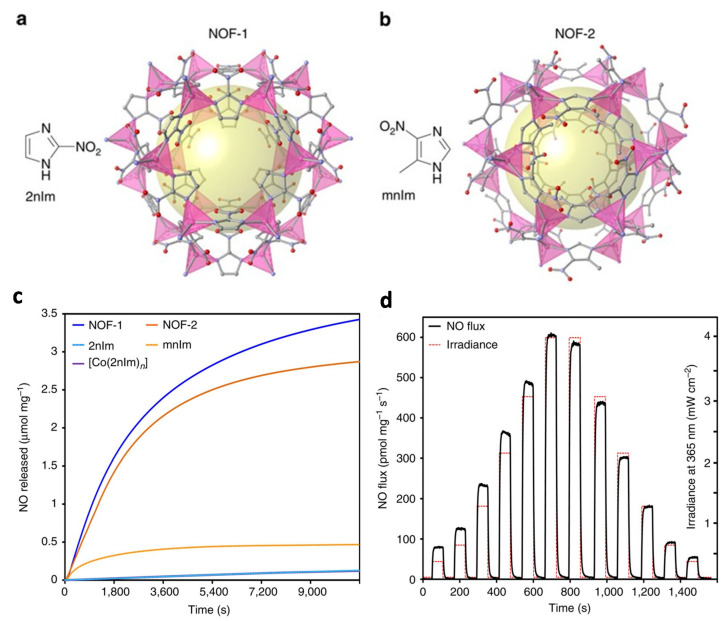
(**a**,**b**) Schematic representation of the NO-releasing frameworks, NOF-1 and NOF-2. (**c**) The photoreactivity of the ligands, 2nIm and mnIm, were remarkably enhanced through the formation of NOF-1 and NOF-2. (**d**) The NO flux produced upon the photoactivation of NOF-1 can be tuned by varying the irradiation conditions. Reproduced with permission from [39], copyright 2013, Nature Publishing Group.

**Figure 6 nanomaterials-10-01134-f006:**
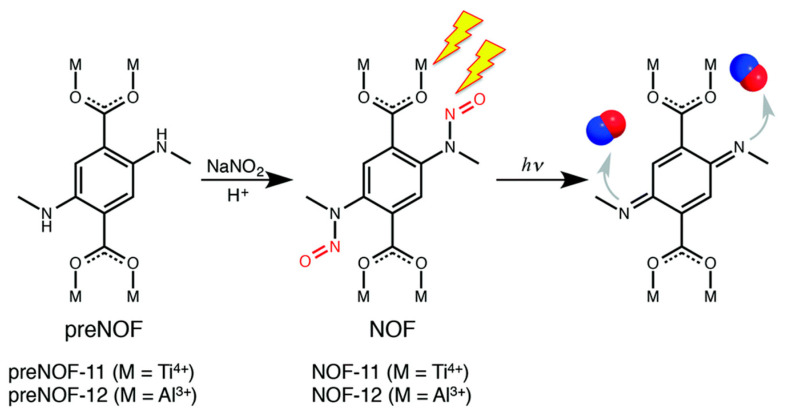
Scheme of conversion by postsynthetic nitrosation and light-induced NO releasing. Reproduced with permission from [40], copyright 2015, the Royal Society of Chemistry.

**Figure 7 nanomaterials-10-01134-f007:**
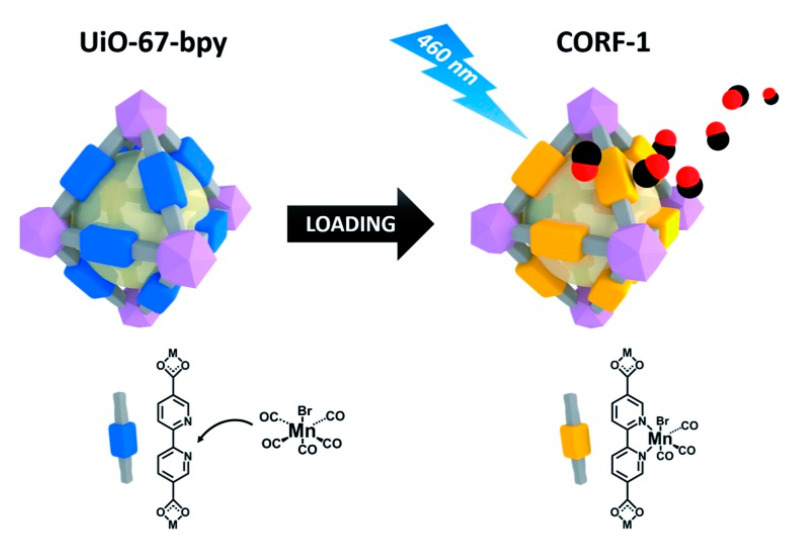
Schematic showing the loading of MnBr(bpy)(CO)_3_ on UiO-67-bpy to synthesize CORF-1, and the subsequent CO release upon light irradiation. Reproduced with permission from [68], copyright 2017, the Royal Society of Chemistry.

**Figure 8 nanomaterials-10-01134-f008:**
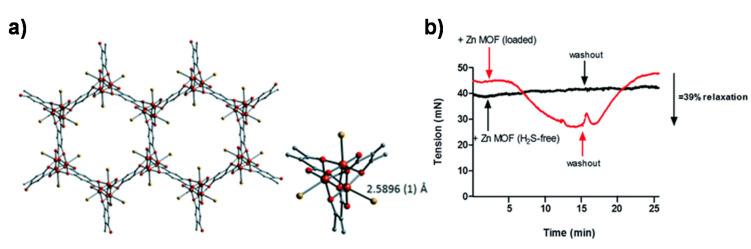
(**a**) H_2_S-loaded Ni-MOF structure from Rietveld refinement of X-ray diffraction data. The metal cluster of Ni-CPO with H_2_S coordinated to the Ni-site. Color key: Ni = orange, oxygen = red, carbon = grey, sulfur = yellow. Hydrogen atoms are not shown. (**b**) The vasodilatory effect of H_2_S loaded Zn-CPO in a precontracted porcine coronary artery. H_2_S-loaded Zn-CPO-27 showed ~39% relaxation after about 15 min, which was reversed after the MOF was removed using a washout procedure. Reproduced with permission from [72], copyright 2012, the Royal Society of Chemistry.

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
