# Peer review of "Engineering Metal–Organic Frameworks (MOFs) for Controlled Delivery of Physiological Gaseous Transmitters"

_nanomaterials, 2020, doi:10.3390/nano10061134_

Round 1
Reviewer 1 Report
This is a good mini-review focusing on the use of MOFs for the delivery of gaseous nitric oxide, carbon monoxide, and hydrogen sulfide. It covers the progress in this subfield in recent years with updated references. Thus, I recommend the publication of this manuscript after a minor revision. I only have one concern for this review, as shown below.
- How about the chemical stability of MOFs? It has been well known that most MOFs are not stable in water. Even some water-stable MOFs have been developed in the recent ten years, most of them are still not stable in ionic solutions, such as acidic solution, alkaline solutions, and phosphate-based solutions (and both acidic solutions and phosphate-based solutions are quite common in biological systems). Thus, how to select the types of MOFs that show different chemical stability for these specific applications? What are the advantages or disadvantages for a MOF that is stable or unstable in the aqueous solutions mentioned above, for these specific delivery applications? I want to see more discussion and perspective regarding this point from the authors. This information will be beneficial for other researchers who intend to design a new MOFs for these applications.
Author Response
Please kindly find our replies to all the comments in the Cover Letter Reply document.

Reviewer 2 Report
The minireview offers a focus on the application of MOF for the controlled release of three gaseous transmitters, namely NO, CO and H2S.
The topic is quite specific and still scarcely investigated. In any case the matter is potentially interesting and the paper offers a first perspective on it.
-Authors do report only few examples of MOF for delivery of CO and H2S. NO was more investigated and several examples are then reported. In the latter case I suggest to add a table for resuming the results and comparing the performances of MOF with other known delivery systems also by adding a quantitative information with reference of standard values for specific applications.
-The effect of the toxicity of metal ions used has been mentioned by the authors but a better screening should be provided. So the advantages of delivery might be compared with the disadvantages of toxicity in case of MOF decomposition. For instance the stability of selected MOFs in buffer solution could be reported if available
-line 103 Release instead of realize
-line 126 Co2+ instead of CO2+
Author Response

(The authors gave the same response as above.)
